# Metasurface-Assisted Terahertz Sensing

**DOI:** 10.3390/s23135902

**Published:** 2023-06-25

**Authors:** Qian Wang, Yuzi Chen, Jinxian Mao, Fengyuan Yang, Nan Wang

**Affiliations:** 1School of Microelectronics, Shanghai University, Shanghai 200000, China; wangqianyi@shu.edu.cn (Q.W.);; 2Shanghai Key Laboratory of Chips and Systems for Intelligent Connected Vehicle, Shanghai University, Shanghai 200000, China

**Keywords:** THz metasurface, THz sensing, biosensing, biodetection

## Abstract

Terahertz (THz) waves, which fall between microwaves and infrared bands, possess intriguing electromagnetic properties of non-ionizing radiation, low photon energy, being highly sensitive to weak resonances, and non-polar material penetrability. Therefore, THz waves are extremely suitable for sensing and detecting chemical, pharmaceutical, and biological molecules. However, the relatively long wavelength of THz waves (30~3000 μm) compared to the size of analytes (1~100 nm for biomolecules, <10 μm for microorganisms) constrains the development of THz-based sensors. To circumvent this problem, metasurface technology, by engineering subwavelength periodic resonators, has gained a great deal of attention to enhance the resonance response of THz waves. Those metasurface-based THz sensors exhibit high sensitivity for label-free sensing, making them appealing for a variety of applications in security, medical applications, and detection. The performance of metasurface-based THz sensors is controlled by geometric structure and material parameters. The operating mechanism is divided into two main categories, passive and active. To have a profound understanding of these metasurface-assisted THz sensing technologies, we review and categorize those THz sensors, based on their operating mechanisms, including resonators for frequency shift sensing, nanogaps for enhanced field confinement, chirality for handedness detection, and active elements (such as graphene and MEMS) for advanced tunable sensing. This comprehensive review can serve as a guideline for future metasurfaces design to assist THz sensing and detection.

## 1. Introduction

The terahertz (THz) band defined at 0.1~10 THz bridges the microwave and infrared spectra [1]. In the past years, the THz spectrum was also known as the “THz gap” due to the lack of effective sources, as well as modulation and detection devices operating at ambient temperature [2,3]. The development of THz technology is largely restricted, especially compared with the vast advancement of microwave and optical technologies. Only recently, a series of efficient ambient temperature THz wave generators and detectors were developed to fill this gap. Furthermore, the invention and commercialization of terahertz time–domain spectroscopy (THz–TDS) opens the pathway for exploring THz applications [4]. 

The unique position of the THz band provides it with inherent electronics and photonics properties, enabling the THz radiation to realize special features that are elusive for other spectral bands [5]. Therefore, THz waves can be used in a variety of fields, such as medicine, biology, telecommunications, security, and many more. The THz wave has some fascinating characteristics, including fingerprint spectroscopy, non-ionization, high penetration, low photon energy, and molecular signatures, which make it a suitable stimulus source for sensing applications [6], complementing or enhancing conventional X-ray solutions. Different from conventional medical diagnosis approaches, the non-ionizing feature of THz waves enables nondestructive biological tissue detection that is extremely desired for healthcare and biodetection [7]. Besides, the THz frequency is closely correlated with the vibrational frequencies of numerous biomolecules, making it highly sensitive to weak resonance between molecules, such as hydrogen bonds, Van der Waals forces, and non-bonding (hydrophobic) interactions that are not discerned by traditional mid-infrared (MIR) spectroscopy [6]. These characteristics allow new approaches for matter detection and new methods for distinguishing biomolecules, such as proteins, DNA, or RNA. The low photon energy and sensitive water absorption support THz waves to detect cancer cells [8,9,10] and other diseases, such as otitis media. In addition, THz waves can penetrate non-polar materials, such as paper, plastic, and textile materials [11,12] to exhibit optic transparency. This transparency in relation to non-polar bonds combined with the non-ionizing feature enables THz waves suitable for security imaging and detection of biological objects. THz spectroscopy has been demonstrated as a powerful tool for substance detection [13]. The above-mentioned characteristics of THz waves largely illustrate their extensive application fields [14,15,16], especially for label-free [17] and non-contact modalities for sensing and biodetection. 

The wavelength of THz waves is approximately from 30 μm to 3000 μm, which is relatively long compared with infrared radiation [18]. This long wavelength results in low spatial resolution and unreliability when measuring substances with dimensions significantly less than the micrometer scale. The typical size of microorganisms is approximately less than 10 μm, and the range of biomolecules is from 1 nm to 100 nm. Hence, there is a distinct difference between the size of analytes and the wavelength of THz waves. These samples exhibit negligible responses to THz radiation, making the testing unattainable with classic technology. 

Metamaterials as advanced emerging technologies open a new channel to solve the restriction of THz sensing. Metamaterials, using artificially designed structures to resonantly couple with the electromagnetic fields, exhibit effective electric (represented by electric permittivity ε) and/or magnetic (represented by magnetic permeability μ) responses not found in nature [19,20,21,22]. Metamaterials usually consist of periodic subwavelength metal or dielectric structures (meta-atoms), with electromagnetic properties primarily determined by their geometries, sizes, and orientations. Dozens of compelling studies in the electromagnetic field have been reported, such as invisibility cloaks [23] and meta-lenses [24,25]. As the two-dimensional counterpart of bulk metamaterials, metasurfaces effectively avoid the inherent drawbacks of bulky size and fabrication complexities, especially for higher-frequency range. Moreover, metasurfaces can achieve sub-wavelength spatial resolution for enhancing THz detection sensitivity. Metasurfaces also have given rise to unprecedented wavefront manipulating, for instance, planar lenses [24,26], ultrathin invisibility cloaks [23,27], and holograms [28,29]. 

Generally, THz sensing has two main approaches, one is measuring the resonant frequency shift from THz devices originating from the changed ambient refractive index around the metasurface, and the other is measuring the change in the absorption peak in the spectrum due to the existence of the analyte. The THz meta-atom usually consists of a patterned structure with precisely designed geometry, such as splits, to generate strong local field confinement under external excitation. Hence, THz metasurfaces can assist sensors to capture and even amplify small changes generated by weak interaction between the electromagnetic excitation and analyte. The resonant frequency, including single- and multi-resonance modes of THz metasurfaces, is used as a reference to present the sensitivity of sensors. THz metasurfaces have innate resonance frequencies determined by the unique geometry of meta-atoms. These devices present redshift or blueshift when they are deposited with analyte samples. The shift of resonant frequency characterizes the sensitivity of THz sensors. 

The associated reviews mainly focused on introducing various applications of THz technology in biodetection [30,31] or discussing different metasurface-based molecules in biomedical sensing and imaging [7,32], and most of them focused on the optical spectrum [33]. This review summarizes recent metasurfaces-assisted THz sensors based on their sensing mechanism, which are classified in Figure 1. Based on the composed material of metasurfaces, sensing strategies are divided into two parts, that is, THz metasurfaces with passive components and active components. Section 2 contains various THz metasurface-assisted sensors, which are analyzed by their working mechanisms, principles, and modes of sensing in detail. Some common structures and characteristics of THz metasurfaces are emphasized from resonator patterns in relation to chiral properties. Finally, a brief conclusion is presented in Section 3.

## 2. Metasurface-Assisted THz Sensing

There are some crucial characteristics for clearly explaining the performance of a sensor, such as figure of merit (FOM), quality factor (Q-factor), and sensitivity. The FOM of a sensor reflects its selectivity, defined as the normalization of the sensitivity to the full width at half maximum (FWHM) of the resonant dip, that is F=S/FWHM. The Q-factor is a measurement of the resonance sharpness, and it is defined as Q-factor = λ/FWHM, where λ is the resonant wavelength [34]. Metasurface-based THz devices completely contribute to improving the sensitivity of THz sensing. Conventionally, there are two definitions of the sensitivity, S. One is frequency sensitivity S=∆f/Δn, where ∆f is the frequency shift of the resonance peak, and Δn is the bulk refractive index (RI) variation, usually expressed as function of refractive index units (RIU). The other is intensity frequency S=∆I/Δn, where ∆I is the change in resonant intensity [35]. In following discussion, the first definition is decisive measurement. The two factors are the main reasons that affect sensors’ performance, but they are not the only ones. Based on the study of detected analytes, the operating mechanisms of metasurface-based devices are extended to adaptation for THz sensing. To achieve various sensing abilities, conventional passive sensing metasurfaces are equipped with refined geometry shapes, such as nanogaps and chiralities. Active components are incorporated with metasurfaces, enabling the tunability of THz sensing. Various THz sensors with different sensing mechanisms are discussed in this section, from passive to active.

To realize the sensing function of a metasurface, each meta-atom requires the optimization of constituent materials, geometric parameters, resonant modes, and so on. The interaction between meta-atoms and incident waves is decided by the integrated metasurface layout. In the design process, many electromagnetic responses can be taken into consideration, such as amplitude (E), phase distraction (φ), polarization state (p), effective nonlinear coefficient (χ), working wavelength range (λ), and local position (x, y). This process can be conceptually described as a system of equations [36]:(1)$$\begin{array}{c} metasurface\left(t;\lambda ;I;\varphi ;p;\sigma ;l\right)=\left\{\begin{array}{c} E\left(x;y;t;\lambda \ldots \right)\\ \varphi \left(x;y;t;\lambda \ldots \right)\\ p\left(x;y;t;\lambda \ldots \right) \end{array}\right. \end{array}$$
where; tenability (t), efficiency (I), spin angular momentum (SAM) (σ), orbital angular momentum (OAM) (l), etc., are desired optical properties. Various mechanisms correspond to different decisive parameters.

### 2.1. Resonant Structure for Frequency Shift Sensing

Typically, a meta-atom engineered with a metal split can be viewed as an inductive-capacitive (LC) equivalent circuit, forming LC resonance, which is excited by incident electromagnetic waves [37]. The resonant frequency f of the LC circuit is determined by the inductor and the capacitor as $f=1/2\pi \sqrt{LC}$, where C and L are the capacitance and inductance, respectively. The analyte deposited on the structure leads to a frequency shift by changing the capacitance. THz metasurface biosensors are widely applied in detecting biomolecules or microorganisms, such as proteins, antibiotics, and viruses.

In 2013, Park et al. applied the single split-ring-resonator (SRR) to detect fungi and bacteria, as shown in Figure 2a [38]. An obvious resonant frequency shift was observed following the deposition of microorganisms. This label-free sensing of higher sensitivity was more appealing than conventional culture-based detection methods, such as polymerase chain reaction (PCR). Afterward, Yoon et al. [39] exposed samples of other microorganisms, such as molds and yeasts, to similar LC resonant metasurface structures (Figure 2b). They utilized a syringe pump to inject a microbial solution into the fluid channel, enabling the identification of the microorganism type, which is based on its intrinsic properties at an early stage, without complex fluorescent labeling and culturing procedures. The above single-split SRR designs had only one resonance peak of the transmission curve. In 2015, Hu et al. studied THz SRR transmission responses in different polarization directions [40]. For a THz wave incident to the split with parallel polarization, only one resonant peak was generated, while vertical wave polarization provides two resonant peaks. They proposed the equivalent circuit model of SRR high-frequency resonance for the first time, and they also used the SRR to detect 93# and 97# gasoline. In 2015, Xie et al. [41] demonstrated the application of THz biosensors in the field of antibiotic detection, as shown in Figure 2c. The sensitivity was significantly increased due to the THz near-field enhancement caused by the square-shaped metallic extraordinary optical transmission (EOT) structure. This method opens up new application directions for metasurface-based THz biosensors. The above studies show the wide detection of microorganisms, gasoline, and antibiotics through THz metasurface biosensors. The following works have further improved their sensitivity by modifying the resonator structures or materials. Ekmekci proposed a novel THz sensor design, which is based on double-sided SRR (DSRR) topology. It had a smaller electrical size and a higher sensitivity of resonance frequency compared to conventional SRR structures [42]. The basic DSRR sensor topology was to realize different types of sensor functions, depending on the properties of the interlayer medium. Based on the typical SRR structure, Aghadjani et al. added a thin-layer dielectric on top of the metallic ring, forming a hybrid Fano-resonant metallic microstructure array–insulator-metal configuration [43]. The sensitivity was increased by three times compared to the most advanced metamaterial sensor platforms. In 2020, Chen et al. proposed a structure consisting of four SRRs and Cross (FSRRC) [44] by rotating the SRR around the direction of the middle gap. Furthermore, when the symmetrical structure was changed to the asymmetrical one, the latter caused Fano resonance [45], resulting in two new resonant peaks and making it more sensitive. In 2022, Niu et al. applied an innovated structure, which achieved ultra-sensitive detection of the virus SARS-CoV-2. As shown in Figure 2d, a circular SRR, possessing three splits, greatly improved the sensitivity by increasing the number of gaps and realizing precise virus detection. Except the in-plane SRRs, the vertical SRRs out-of-plane also offer strong plasmon resonance with high Q-factor and low Ohmic losses [46]. In 2021, Li et al. manufactured a vertical SRR utilizing three-dimensional printing technology [47]. The maximum value of the Q-factor occurring in the absorption peak at 0.797 THz is 152.9.

### 2.2. Nanogap Enhanced Sensing

Generally, it is hard to trap and detect microorganisms (less than 10 μm) or biomolecules (nano-level) due to their small scales. To improve the sensitivity of LC-resonant meta-atoms for THz sensing, nanogap splits are proposed. A larger capacitance and a stronger field confinement are generated around the nanogap, enhancing sensitivity. Analytes deposited on the nanogap tune the effective dielectric constants of the capacitance, resulting in a resonant frequency shift.

In 2015, Park H.-R. et al. experimentally demonstrated the detection of 1-nm-thick dielectric overlayers using tightly confined THz waves [49], as shown in Figure 3a. The THz waves were squeezed inside sub-10 nm metallic gaps to detect refractive index changes caused by a thin dielectric overlayer. The measurement indicated that it was possible to detect even sub-1-nm dielectric overlayers. Besides refractive index sensing, the THz nanogap platform was also utilized for surface-enhanced THz absorption. Based on these findings, detecting such thin dielectric overlayers using the significant field enhancement at nanoscale gaps has potential applications in various fields. In 2017, Park S. J. et al. demonstrated the use of a nanogap–metasurface-based THz sensor for virus detection [50]. They applied SRRs (Figure 3b) with various gap widths (Figure 3c) for the highly sensitive detection of viruses. Two types of viruses, PRD1 and MS2, were tested on the metasurface at low densities. This sensing test indicated that reducing the gap width increased sensitivity by up to 13 times. In the same year, Shukla et al. proposed a biosensor [51] with nanogaps formed by graphene and hexagonal boron nitride for DNA sequencing. This graphene–hBN heterostructure served as a useful nanodevice for sensing applications, which may promote the development of nanodevice DNA sensors [52]. In 2021, Chen changed the structure size of the FSRRC to the nanometer level for transmission detection [53]. Two frequency signatures make it ideal for the accurate identification of small pathogens, such as viruses. In 2023, Ji et al. presented a highly sensitive method for detecting viruses using gold nanogaps filled with Al_2_O_3_ [54]. They demonstrated that the resonant frequency of these gaps changes significantly when target viruses are deposited on them, allowing for low-virus-density detection. Therefore, SRR engineered with nanogaps of appropriate geometry size increased sensitivity in virus detection. However, in terms of fabrication, nanogap sensors confront the difficulties of lithography technology in practical applications [55]. Recently, atomic layer deposition (ALD) technology [56], capable of solving the challenge, has been developed for fabricating nanogaps with a width of less than 1 nm.

### 2.3. Chirality for Polarization Sensing

Chirality refers to the non-superposability of an object on its mirror image. A chiral object and its mirror image are called enantiomers, as shown in Figure 4a [57]. Since structural enantiomers are very different in properties, the chiral phenomenon plays a key role in chemical, biological, and physiological processing, making chirality a useful tool for identifying molecules in pharmaceutics, toxicology, and stereochemistry [58]. Chiral substances exhibit different refractive indices to asymmetric electromagnetic fields excited by different circularly polarized waves. The refractive index is usually a complex number, in which the imaginary parts of the left and right circularly polarized (LCP and RCP) excitations are different and lead to different absorption rates. This circular dichroism (CD) phenomenon and optical rotatory dispersion (ORD), which are also called optical activity, are manifested by a difference in intensity and phase responses between LCP and RCP light illuminations through the chiral medium [59]. This is because linearly polarized light can be regarded as a superposition of orthogonal LCP and RCP waves. A different real part of the refractive index produces a phase difference of these two orthogonal circularly polarized waves, which rotates the linearly polarized wave. The modulation of circularly polarized light is achieved by the Pancharatnam–Berry (PB) phase [60].

Utilizing chirality for sensing has been demonstrated in the visible light spectrum, providing selectivity for molecules with handedness [61,62,63]. The development of THz technology brings chirality to a lower frequency. Circular dichroic spectroscopy and optical rotation spectroscopy have been widely used to identify the special arrangement of substances in biological [64], chemical [65], and physical processes [66]. Effective detection of the substances’ structure has important applications in the microscopic field. However, the chirality and electromagnetic responses of most natural chiral media are very weak. Thus, chiral metamaterials and chiral metasurfaces have been proposed, enhancing the optical rotation and circular dichroism responses to orders of magnitude higher, thus meeting more needs in different fields.

In 2020, a three-dimensional chiral structure, consisting of graphene, dielectric, and metal in the THz region was proposed, as shown in Figure 4c [67]. This biosensor device had a good sensing characteristic in chiral biomolecules (collagens). In the same year, a double-layer chiral metasurface was designed by Zhang et al. [68]. The special symmetric geometry of each meta-atom, with its adjacent cells, provided a strong chiral electromagnetic response, leading to a strong polarization conversion. Moreover, compared with the traditional THz transmitted resonance sensing for film thickness, polarization sensing was characterized by polarization elliptical angle (PEA) and polarization rotation angle (PRA), with a better Q-factor and FOM. The chiral metasurface and its polarization sensing method provide new methods for high-efficiency THz polarization manipulation and high-sensitive sensing by using THz polarization spectroscopy. In the next year, they demonstrated accurate measurement of chiral enantiomers of three amino acid aqueous solutions [64]. Shi et al. [69] demonstrated a THz all-dielectric metasurface with crescent cylinder arrays for chiral drug sensing, as shown in Figure 4b. It was able to monitor the concentrations of ibuprofen solutions, and the maximum sensitivity reached 60.42 GHz/mg. Furthermore, the metasurface successfully distinguished different chiral molecules, such as (R)-(−)-ibuprofen and (S)-(+)-ibuprofen, in the 5 µL trace amounts of samples. Zheng et al. proposed an all-silicon chiral metasurface for highly efficient circular polarization differential transmittance in the THz band [70]. Many potential applications in photon-spin selective devices were found, such as circularly polarized light detectors and chiral sensors. In 2022, Zhang et al. reported a spatial symmetry-breaking chiral THz metamaterial structure with stacked layers of L-shape arranged gold disks as the periodic meta-atom [71]. The device realized label-free detection of proline in biological samples and label-free enantio-discrimination of chiral molecules.
Figure 4Chirality structures for polarization and handedness sensing at THz frequencies. (**a**) Schematic of chiral molecules. Reprinted (adapted) with permission from [57]. Copyright 2015 American Chemical Society. (**b**) A schematic illustration of the THz chiral metasurface sensing experiment. Reprinted from Ref. [69]. (**c**) A periodic bi-layer three-dimensional THz graphene metasurface with a chiral structure. Reprinted from Ref. [61].
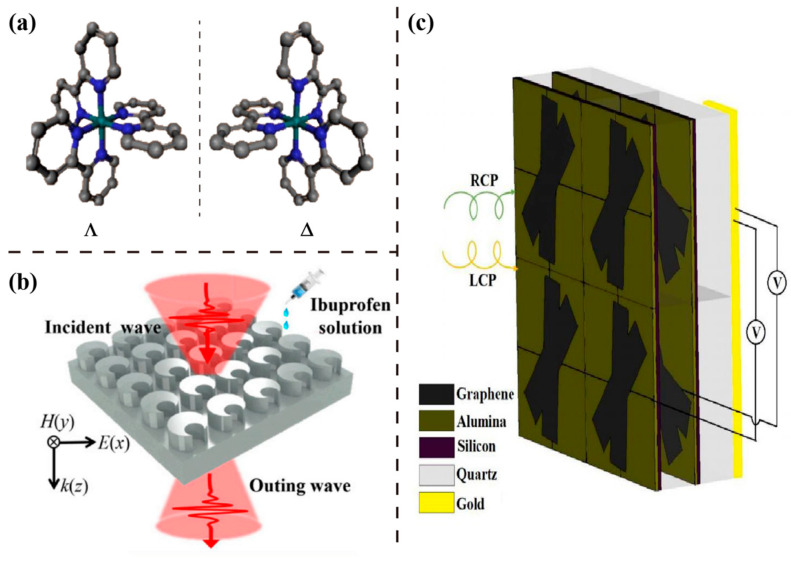


### 2.4. A Metasurface with Active Components for THz Sensing

Conventional metasurfaces with fixed structures limit the functionality of THz wavefront modulation. Once the metasurface is fabricated, it manipulates THz waves in a static pre-defined manner. The lack of tunability restricts the passive metasurface detection to no more than two different samples. Quasi-static inspection in previous research was barely adapted to a complex environment. In addition, it is difficult to satisfy the advanced requirements for high intelligence and a self-regulated system in future applications. Tunable metasurfaces incorporated with active materials or MEMS structures realize dynamic manipulation by adjusting the material properties or the meta-atoms’ geometry under external excitations [72]. Metasurfaces with active components have been studied by researchers in recent years, such as phase-change material (GST and VO_2_) [73,74], active electronics (diodes) [75,76], two-dimensional material (graphene) [77], perovskites, and MEMS [78]. The function of active metasufaces that have high flexibility for manipulating electromagnetic waves is tunable and even reconfigurable. The concept of reconfigurable intelligent surfaces (RISs) composed of versatile active metasurfaces has been proposed and researched in many areas, such as secure satellite transmission networks [79,80,81,82]. Especially, graphene and MEMS are suitable for metasurface-based THz sensing due to their tunable characteristics, effectively functioning on the THz band, leading to a research hotspot in the biodetection field during the decade. The THz biosensors exhibit optimal performance by tuning active components, and they can even distinguish two analytes in the same structure.

#### 2.4.1. Graphene-Based THz Sensing 

The advancement of biodetection is necessary for preventing the spread of disease and providing effective medical treatment. Owing to the presence of π-electrons, graphene has a great affinity for biomolecules and shows excellent potential for sensing small and organic macromolecules [83]. Thus, graphene stands out from massive active materials due to its bio-friendly and tunable conductivity. The THz metasurface with a graphene element exhibits a kind of inoffensive detection for the human body in medical diagnosis [84,85]. Graphene is an ultrathin two-dimensional material consisting of a monolayer carbon atom. This special atomic structure grants graphene excellent mechanical and electromagnetic properties. The surface conductivity of graphene is artificially modulated by applying bias voltage or chemical doping to change its carrier concentration [86]. The theoretical model of graphene’s surface conductivity $\sigma_{g}$ is established by the Kubo formula [87], including intra-band and inter-band carrier transition in the entire electromagnetic spectrum. According to the Pauli Exclusion Principle [88], due to the photon energy being less than $E_{f}$, the intra-band part becomes dominant, and the inter-band, consequently, is neglected in the THz regime. Thus, the surface conductivity of graphene $\sigma_{g}$ is described by the Drude-like model [89] as:(2)$$\sigma_{g}=\frac{ie^{2}E_{f}}{\pi \hbar^{2}\left(\omega +i/\tau \right)}$$
where; e is the electron charge, ω is the angular frequency of the incident wave, ℏ is the reduced Planck’s constant, τ is the relaxation time of graphene, and $E_{f}$ is the Fermi energy of graphene. It indicates that the surface conductivity of graphene can be modulated by the Fermi level. Meanwhile, graphene is an excellent candidate for replacing conventional metal structures due to its local surface plasmon resonance in the THz regime. The external excitation is applied to graphene to tune the Fermi level, and the surface conductivity of graphene is subsequently affected. Traditionally, chemical vapor deposition (CVD) is a predominant method to prepare graphene flakes [90]. However, it is inevitably affected by impurities, defects, and disorders. CVD-graphene is usually p-doped, so the initial Fermi level $E_{f}$ slightly deviates from the Dirac point and is in the valence band [91], namely, the initial Fermi level of CVD-graphene is quite close to the Dirac point. An extremely weak excitation can change the initial Fermi level from the valence band to the Dirac point, exhibiting excellent traits as an ultra-sensitive biosensor. 

It is simple to create a large frequency shift or an apparent amplitude change for graphene-based plasmonic devices. Initially, the typical SRR structures [92,93] of graphene-based metasurfaces were designed to sense the frequency shift. In 2021, Amin et al. presented a chiral graphene plasmonic metasurface to detect various viruses by the reflected polarization state of analytes in the THz spectrum [94]. A diagonally symmetrical L-shaped graphene split ring was chosen as the chiral resonant element to detect analytes with different refractive indices, as illuminated in Figure 5a. Compared with no graphene, the structure with graphene induced stronger interaction between the target biomolecules and the metasurface. Three viruses, including H1N1, H5N2, and H9N2, were measured, exhibiting different polarization ellipse orientations of −3°, 36°, and 48° for H9N2 (left handed), H1N1 (right handed), and H5N2 (right handed), respectively, at 1.364 THz.

Owing to the high security, reliability, and sensitivity of graphene-based metasurfaces, a series of graphene-based THz sensors were investigated [95,96,97,98]. In 2022, Hu et al. presented an original structure of the metasurface sensor, consisting of six layers, as shown in Figure 5b [99]. The absorption peaks of the proposed sensor depended on the thickness of the air layer. There was a good linear relationship between the absorption peak and dielectric parameters of the analyte. When the impedance of the air layer was matched with the equivalent impedance of the proposed sensor, the performance of absorption was the best for sensing. By adjusting the Fermi level of the full structure, the optimal sensitivity was 0.45 THz/RIU in the range of 0.1~2 THz for the analyte placed at the air layer (Figure 5c). 

Except for using the bare patterned graphene for THz sensing, a metal–graphene hybrid THz metasurface was also presented by Sun et al. [100]. Compared with the bare metal metasurface, introducing graphene improved the sensitivity of metasurface sensors to a great extent. The proposed metal–graphene hybrid metasurface (MGHM) shown in Figure 5d consisted of a classical plasmon-induced transparency (PIT) metal metasurface with monolayer graphene for detecting nortriptyline. The PIT resonator was a classical structure to modulate the transmitted THz waves using the Fermi level and is capable of generating the PIT phenomenon for sensing (Figure 5e) [101,102,103]. Thus, the evaluation criteria of sensitivity were to calculate the value of ∆T, which was defined by the difference between the peak and the valley of the transmission spectrum. Because the field confine was magnified by the strong interaction between graphene and PIT, the experimental results of ∆T reached 55.3% of MGHM, which was enhanced 3.4-fold with 1 ng nortriptyline analyte compared with the metal metasurface, and the minimum detection limit was extended down to 0.1 ng. Similarly, Yan et al. proposed a metal–graphene hybrid metasurface [104]. The proposed sensor added perovskites to the graphene structure, making the Fermi level of graphene p-doped closer to the Dirac point, namely, the proposed structure had an ultra-high sensitivity, operating in the THz band. Then, sericin deposited on the hybrid metasurface moved the valence band to the Dirac point with a low concentration. The metal component of the biosensor possessed the characteristic of electromagnetically induced transparency (EIT). This EIT-like metasurface detected sericin with a detection limit of 780 pg/mL.
Figure 5Graphene-based metasurfaces for THz sensing. (**a**) Spectrum-free (monochromatic) polarization state sensitive biosensing with linearly *x*-polarized electric field impinges on dispersion-engineered chiral LSPR metasurface and three different reflected signals of the left (R_1_)- and right (R_3_) handed elliptical and linearly *y*-polarized (R_2_) waves. Reprinted from Ref. [94]. (**b**) Layered structure diagram of the graphene-based THz metasurface sensor and (**c**) the absorption spectra corresponding to the Fermi level (Ef) from 0 to 1.0 eV. Reprinted from Ref. [99]. (**d**) Schematic of THz transmitted detection on MGHM sensor and (**e**) the corresponding simulated transmission spectra with different Fermi levels of graphene. Reprinted from Ref. [100].
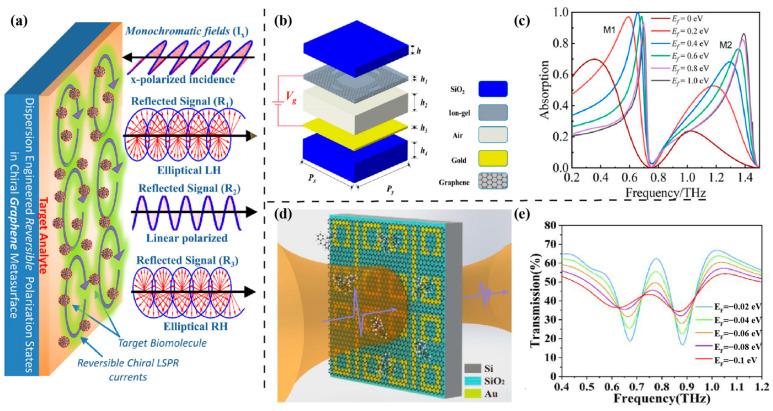


#### 2.4.2. MEMS-Based THz Sensing

Advanced MEMS technology is applied for sensing [105]. It is greatly conducive to realizing mechanical manipulation on a nano-scale with high-speed and enhanced flexibility. The micro-fabrication processes, based on photolithography, provide high precision for MEMS-based modulation [106]. Two tunable mechanisms are principally studied in the field of MEMS-based metasurfaces and are presented in the following subsection. One involves utilizing the difference in the thermal expansion coefficient of bi-materials to bend the cantilevers after the release process in fabrication, and the other is modulating the geometrical dimension of MEMS-based devices actuated by the electrostatic force. MEMS-based sensors have been demonstrated in optical sensing [107]. Besides, a myriad of MEMS-based metasurfaces used in the THz range has been deeply studied in the past decade.

In 2013, Alves et al. first presented a structure of a MEMS-based absorber incorporated with two cantilevers, as shown in Figure 6a [108]. The cantilever produced an angular displacement, measuring the responsivity of the MEMS-based sensors. The angular deformation of the sensor changed with increased temperature (Figure 6b), and the maximum measured value was 1.2 deg/μW at 3.8 THz. The proposed MEMS metasurface demonstrated potential for sensing. However, the bi-material cantilever had inherent angular deflection caused by the residual stress after MEMS fabrication. To solve the problem, Alves et al. subsequently introduced one additional cantilever on each side of the structure to compensate for the stress [109]. The self-leveling sensor eliminated the undesirable effect. Then in 2022, to overcome the single and narrowband response of the sensors, Ozpinar designed a novel structure in ref. [110], which was a fractal metasurface absorber actuated by MEMS thermomechanical bi-material microcantilevers. The unit cell consisted of triangular sectional fractal metasurfaces, shown in Figure 6e, resulting in penta-band absorption at 1.1 THz, 3.4 THz, 4.9 THz, 5.9 THz, and 7.8 THz, respectively. All five absorption peaks were more than 95%, up to 45°, corresponding to five bandwidths of 35.5 GHz, 114.0 GHz, 131.9 GHz, 146.1 GHz, and 264.2 GHz, respectively. This enabled THz sensors to cover broadband response.

In 2021, Zhong et al. [111] presented a structure of a MEMS-based tunable terahertz metamaterial (TTM) composed of an inner triadius and outer electric split-ring resonator (eSRR) structures, as shown in Figure 6c. It was similar to ref. [48], but the height of the inner triadius was not fixed. Meanwhile, Yang et al., from the same group, also published a similar structure [112], consisting of two concentric split-ring resonators (SRRs), as shown in Figure 6f. The attractive point was that the height h between SRRs and the substrate was precisely regulated by MEMS technology in these two structures. Based on the electromagnetically induced transparency (EIT) of TTM, the peak of resonant absorption resulted in a big shift with the change in ambient refractive index. The Q-factor value of resonances were 66.01 for Zhong and 50.69 for Yang, corresponding to sensitivities of 0.379 THz/RIU and 0.54 THz/RIU, respectively. Although the Q-factor of Zhong was higher than that of Yang, the calculated value of sensitivity was inverse because the structure of the latter could induce dipole resonance coupling with LC resonance, leading to the enhancement of the field response. These researches proved the practicability of MEMS-based sensors with high sensitivity, reliability, and security for THz sensing.

### 2.5. Non-Typical Metasurface for THz Sensing

In addition to the metasurface-assisted THz sensing mechanisms discussed above, the emerging metasurface tunable method through microfluid [113] also indicates a high potential. In 2021, Fan et al. utilized ferromagnetic nanofluid combined with a chiral metasurface to enhance chirality detection sensitivity based on the magneto-optical effect for sensing nanoparticles [114]. In 2022, Mu et al. experimentally demonstrated that the deposition of liquid silver nanoparticles on a THz metasurface enhanced the local electric field and augmented the substance detection sensitivity by two orders of magnitude [115]. 

Furthermore, except for the abovementioned structural innovated method, a selective substrate is an alternative approach for effective THz sensing. Tao et al. found that using a high dielectric constant substrate, such as gallium arsenide and high-resistance Si, induced a frequency shift of 19.3% (3.9% for a general substrate) [116]. This result indicates that adjusting the substrate material is another promising technique for improving sensor performance.

## 3. Conclusions

In conclusion, this review provides a comprehensive investigation of metasurface-assisted THz sensors, including their structure characteristics, working principles, potential applications, and corresponding sensing performances. Additionally, representives of the sensing performances are summarized in Table 1. The metasurfaces were categorized and analyzed according to the THz sensing mechanisms, which mainly depend on using various resonator structures with diverse geometries, dimensions, and materials to realize different sensing applications, such as in medical applications and security. Compared with conventional sensors, the unique characteristics of the metasurface provide unparalleled facilitation in superior sensitivity, greater stability, and label-free safety. Experiments and numerical simulations of the aforementioned studies demonstrated excellent sensing performance. In particular, active metasurfaces based on graphene or MEMS technology have great flexibility and compatibility in detection, and the performance of sensing can be tuned to the maximum value under a low concentration of the analyte. Metasurfacce-based THz sensors have significant potency in microorganism sensing and cancer detection with a nondestructive and non-contact way, which will drive bio-detection to be more secure in the future. Additionally, the relevant manufacturing technologies, such as atomic layer lithography, are rapidly evolving to hold the promise of fabricating mass sensors. Hence, further investigation of metasurface-based THz sensing will lead to technological breakthroughs in the multidisciplinary scientific area, contributing to the advancement of sensing, as well as biodetection industries.

## Figures and Tables

**Figure 1 sensors-23-05902-f001:**
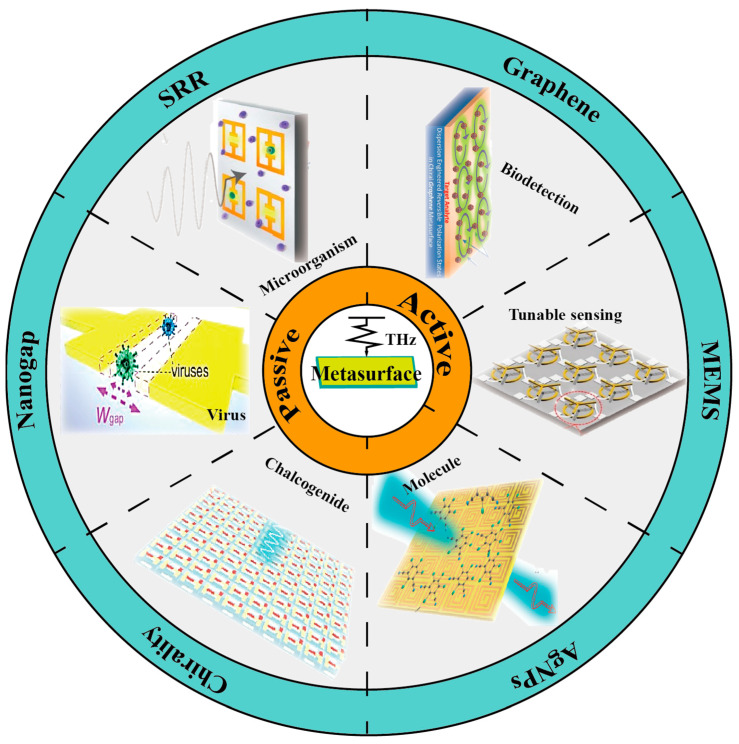
A schematic illustration of various metasurface-assisted THz sensors.

**Figure 2 sensors-23-05902-f002:**
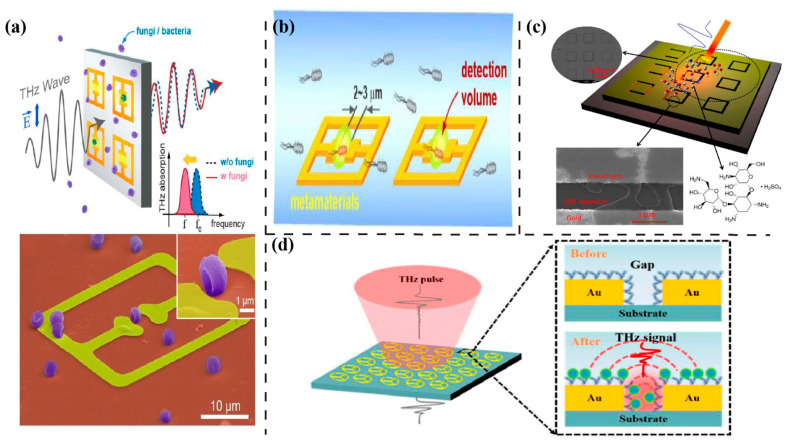
Resonant structures for frequency shift sensing. (**a**) Schematic and SEM image (bottom) of the THz metamaterial for microorganism sensing. Reprinted from Ref. [38]. (**b**) THz metasurface sensing for liquids containing low-density microorganisms. Reprinted from Ref. [39]. (**c**) Schematic and SEM images of the metasurface structure deposited with kanamycin sulfate molecules on top. Reprinted from Ref. [41]. (**d**) A schematic illustration of the proposed three-split-ring (TSR) resonator with a magnification of the gap area. Reprinted from Ref. [48].2.2. Nanogap Enhanced Sensing.

**Figure 3 sensors-23-05902-f003:**
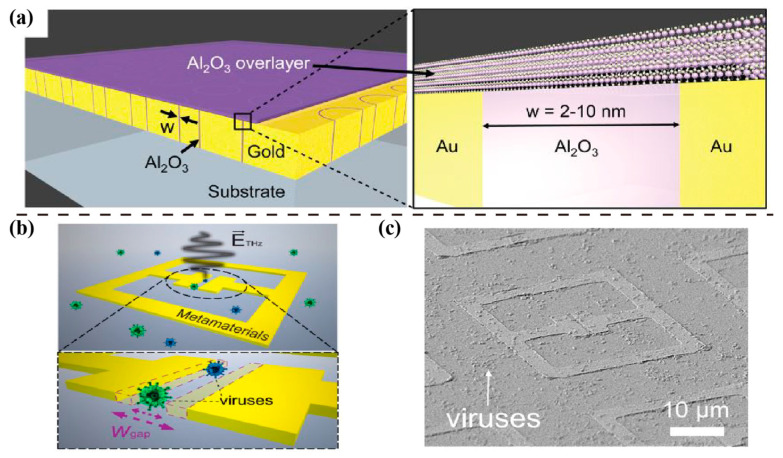
Resonant structures with nanogaps for local field enhanced sensing. (**a**) Schematic of an ALD-deposited ultrathin Al_2_O_3_ layer on an annular gap array, with an enlarged view of the nanogap portion. Reprinted from Ref.. (**b**) THz nanogap metasurface for viruses sensing and (**c**) SEM image of viruses deposited on the nanogap (width of 200 nm) metasurface sensor. Reprinted from Ref.

**Figure 6 sensors-23-05902-f006:**
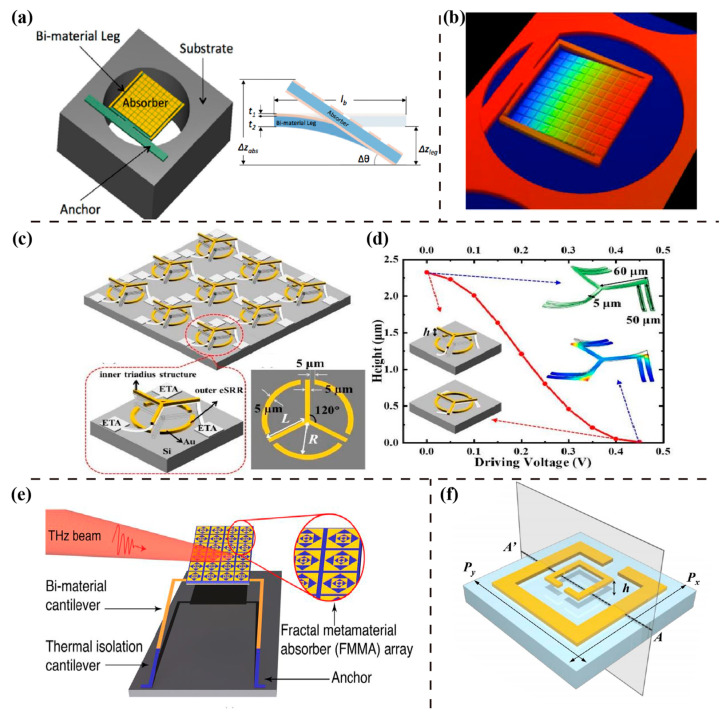
MEMS-based metasurfaces for THz sensing. (**a**) A THz bi-material sensor with metamaterial absorber fabricated on a Si substrate and the schematic of amplified cantilever and (**b**) its three-dimensional optical profile. Reprinted from Ref. [108]. (**c**) A schematic illustration of a MEMS-based tunable THz metasurface, which is controlled by different deriving voltages, as well as (**d**) its elevating height as a function of the driving voltages. Reprinted from Ref. [111]. (**e**) A FMMA-based sensor. Reprinted from Ref. [110]. (**f**) The unit cell structure of a MEMS-based tunable THz metasurface. Reprinted from Ref. [112].

**Table 1 sensors-23-05902-t001:** Sensing performance comparisons of various mechanisms based on THz metasurfaces.

Sensing Mechanism	$n_{a}$	∆f/∆I (GHz)	S (GHz/RIU)	FOM	Q-Factor	Analyte	Concentration	Ref.
SRR	1.37	9	24.32	0.1216	5.58	Penicillia	$0.090\;{\mu m}^{-2}$	[38]
Nanogap	-	40	-	-	-	Al_2_O_3_	1 nm thick	[49]
	3.48/3.83	60/69	24.20/24.38	-	-	Virus	$4\;{\mu m}^{-2}$	[50]
Chirality	1.15	144 (CD)	960	-	-	Collagen	1.5 nm thick	[67]
	-	181.25	-	-	-	Ibuprofen	0.3 mg/mL ×10	[69]
Graphene	2.0	0.203 (0.377)	203 (377)	1.81 (1.57)	8.21 (6.05)	-	-	[99]
	-	42	-	-	-	Nortriptyline	1 ng	[100]
MEMS	1.42	160	379	71.33	72.47	-	-	[111]
	1.1	54 (112)	540 (1120)	57.4 (44.7)	50.7 (40.0)	-	-	[112]
Microfluid	1.6	170	283.3	-	-	Polystyrene particle	-	[113]

$n_{a}$ is the refractive index of analyte.

## Data Availability

Not applicable.

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
