# Peer review of "Metasurface-Assisted Terahertz Sensing"

_sensors, 2023, doi:10.3390/s23135902_

Round 1
Reviewer 1 Report
The authors have provided an insightful review on the recent advancements in THz Metasurfaces for sensing purposes, with a particular focus on their mechanism, in their manuscript titled "Metasurface-Assisted Terahertz Sensing." The breadth of topics covered in a concise manner is impressive. This review will be valuable to researchers in this field and also to future scientists and engineers who are interested in pursuing research in this area. Therefore, I recommend the publication of this review after a minor revision.
However, in section 2.1, the authors have primarily discussed in-plane split-ring-resonators (SRRs), but they have not mentioned vertical SRRs at all. It is recommended that the authors incorporate several sentences discussing the advancements in vertical SRRs, including works such as [Liang, Yao, et al. "Bound states in the continuum in anisotropic plasmonic metasurfaces." Nano Letters 20.9 (2020): 6351-6356; Li S, Zhang L, Chen X. 3D-printed terahertz metamaterial absorber based on vertical split-ring resonator, Journal of Applied Physics, 2021, 130(3): 034504].
Furthermore, in section 2.3, the authors' explanation of the difference between circular dichroism (CD) and optical rotatory dispersion (ORD) was not very clear, and some information provided was misleading. The authors should double-check the definition of circular dichroism since it often has different definitions, not simply based on the absorption difference between left-circularly polarized (LCP) and right-circularly polarized (RCP) light. For example, [Shi T, Deng Z L, Geng G, et al. Planar chiral metasurfaces with maximal and tunable chiroptical response driven by bound states in the continuum[J]. Nature Communications, 2022, 13(1): 4111] provides more clarity on the topic.
readable
Reviewer 2 Report
Please see the attached file.

Minor editing of English language is needed.
Reviewer 3 Report
This manuscript (sensors-2462071; Metasurface-Assisted Terahertz Sensing) is a condensed review to address recent advances in metasurface-based advanced techniques by tuneable sensing using terahertz frequency (from microwave to infrared wave spectra). Metasurface could be utilized in biodetection, biomedical sensing, and biomedical imaging.
Below are comments that could be considered before publication.
1. Theoretical aspects.
(1) Please support more explanation on terms such as refractive index unit (RIU), Figure of merit (F or FoM), quality factor (Q), and detection accuracy. I believe the definition is like the one elsewhere (example: Adv. Optical Mater. 8, 2000865 (2020), Biomedical Optics Express, 11(5), 2416-2430 (2020).).
(2) Sensitivity has a 2-definitions in terms of the RIU unit. First is frequency sensitivity (? = ∆?/∆?), and second is intensity sensitivity (? = ∆I/∆?). These should be separately described in the manuscript. (example: Frontiers in Physics, 622 (2021)).
(3) What could be the origin of the ∆? or ∆I in devices?
(4) Could you provide a few theoretical aspects of meta-atoms (resonator structure)? Structural considerations could be enough. (Following publications might be helpful: Integrated-resonant metadevices: a review. Advanced Photonics, 5(2), 024001-024001 (2023)., A review on metasurface: from principle to smart metadevices. Frontiers in Physics, 8, 586087 (2021).)
2. Informative aspects.
(1) Could you summarize the performance of the mentioned results? The author might consider the structure in Table 1 of Adv. Optical Mater. 8(3), 1900721 (2020).
(2) It could be more informative if some examples with non-graphene materials are added.
(3) How about considering the aspect with reversibility or reusability of sensor?
(4) How about the performance of the metasurface THz sensor to other competitive technology?
3. Typo
(1) CAD should be CVD. ”Traditionally, chemical vapor deposition (CAD)”.
4. Perspective
(1) What direction would the author expect specifically for future directions or developments on metasurface-based THz sensors?
I believe this is a well-written review.
